# A Diagnostic Gene Expression Signature for Bladder Cancer Can Stratify Cases into Prescribed Molecular Subtypes and Predict Outcome

**DOI:** 10.3390/diagnostics12081801

**Published:** 2022-07-25

**Authors:** Runpu Chen, Ian Pagano, Yijun Sun, Kaoru Murakami, Steve Goodison, Ramanathan Vairavan, Malak Tahsin, Peter C. Black, Charles J. Rosser, Hideki Furuya

**Affiliations:** 1Department of Microbiology and Immunology, The State University of New York at Buffalo, Buffalo, NY 14260, USA; runpuche@buffalo.edu; 2Cancer Prevention and Control Program, University of Hawaii Cancer Center, Honolulu, HI 96813, USA; pagano@hawaii.edu; 3Department of Computer Science and Engineering, The State University of New York at Buffalo, Buffalo, NY 14260, USA; yijunsun@buffalo.edu; 4Department of Biostatistics, The State University of New York at Buffalo, Buffalo, NY 14260, USA; 5Cedars-Sinai Medical Center, Samuel Oschin Comprehensive Cancer Institute, Los Angeles, CA 90048, USA; kaoru.murakami@cshs.org (K.M.); charles.rosser@cshs.org (C.J.R.); 6Quantitative Health Sciences, Mayo Clinic, Jacksonville, FL 32224, USA; goodison.steven@mayo.edu; 7DiaCarta Inc., Pleasanton, CA 94588, USA; rvairavan@diacarta.com (R.V.); malak@diacarta.com (M.T.); 8Department of Urologic Sciences, University of British Columbia, Vancouver, BC V6T 1Z4, Canada; peter.black@ubc.ca; 9Department of Biomedical Sciences, Cedars-Sinai Medical Center, Los Angeles, CA 90048, USA; 10Nonagen Bioscience Corp., Los Angeles, CA 90010, USA

**Keywords:** bladder cancer, biomarker, molecular diagnostic, outcomes, multiplex

## Abstract

Bladder cancer is a biologically heterogeneous disease with variable clinical presentations, outcomes and responses to therapy. Thus, the clinical utility of single biomarkers for the detection and prediction of biological behavior of bladder cancer is limited. We have previously identified and validated a bladder cancer diagnostic signature composed of 10 biomarkers, which has been incorporated into a multiplex immunoassay bladder cancer test, Oncuria™. In this study, we evaluate whether these 10 biomarkers can assist in the prediction of bladder cancer clinical outcomes. Tumor gene expression and patient survival data from bladder cancer cases from The Cancer Genome Atlas (TCGA) were analyzed. Alignment between the mRNA expression of 10 biomarkers and the TCGA 2017 subtype classification was assessed. Kaplan–Meier analysis of multiple gene expression datasets indicated that high expression of the combined 10 biomarkers correlated with a significant reduction in overall survival. The analysis of three independent, publicly available gene expression datasets confirmed that multiplex prognostic models outperformed single biomarkers. In total, 8 of the 10 biomarkers from the Oncuria™ test were significantly associated with either luminal or basal molecular subtypes, and thus, the test has the potential to assist in the prediction of clinical outcome.

## 1. Introduction

Bladder cancer is a biologically heterogeneous disease with variable clinical presentation, response to therapy and clinical outcome. The molecular complexity of bladder cancer has restricted the clinical utility of tests that rely on single features or biomarkers for the detection and prediction of bladder cancer behavior [1,2,3]. The emergence of high-throughput molecular profiling technologies has enabled the development of multiplex molecular signatures [4,5] with potential use for diagnosis, staging, prognostication and therapeutic decision making. There are currently two FDA-approved multiplex molecular tests for bladder cancer, UroVysion and the Immunocyt/Ucyt + Test, but their clinical utility has been impacted by limited sensitivity and specificity [6]. The next generation of promising multiplex tests in the field include Oncuria™, CxBladder™, XPert^®^ Bladder Cancer Test, AssureMDx^®^ and UroSEEK.

Oncuria™ is a multiplex immunoassay that quantitatively monitors a bladder cancer-associated diagnostic signature composed of 10 protein biomarkers (ANG, APOE, A1AT, CA9, IL8, MMP9, MMP10, PAI1, SDC1 and VEGFA) [7]. In a series of studies, the molecular signature was developed and tested for the non-invasive detection of bladder cancer through urinalysis [8,9,10,11,12,13]. In addition, immunostaining studies in excised bladder tumor tissues showed that expression of the Oncuria™ biomarkers was increased in neoplastic over benign urothelium and high levels were associated with reduced overall patient survival [14,15].

The molecular subtyping of a range of solid tumors has emerged as a valuable tool for the classification of patients into genetically homogenous groups to guide clinical management [16]. A number of subtyping schemes have been proposed for bladder cancer [17,18,19,20,21] with varying levels of complexity. While meta-analyses and comparisons across cohorts have begun to reveal some common trends within the classification systems [22], significant heterogeneity within individual subgroups remains and more work is needed in order to define a unified classification system that can gain wide acceptance. Herein, we analyzed a series of gene expression datasets from TCGA and the Gene Expression Omnibus (GEO) to evaluate the potential utility of the biomarkers comprising the Oncuria™ test signature for the molecular subtyping of bladder cancer and the prediction of clinical outcome.

## 2. Materials and Methods

### 2.1. Data Acquisition

A discovery cohort was composed of 430 samples from TCGA with gene transcriptome data of which 404 patients had valid survival data (19 normal and 411 cancer). The dataset includes only one non-muscle invasive bladder cancer (NMIBC) with the rest being muscle invasive bladder cancer (MIBC) patients. Three additional datasets were accessed for validation analyses: GSE87304, including 303 MIBC patients with the primary outcome of recurrence free survival [23], GSE48075, including 142 NMIBC patients with the primary outcome of overall survival [19], and GSE32894, including 215 NMIBC and 93 MIBC patients [18] patients with the primary outcome of disease specific survival, respectively. These datasets are an open resource with no noted ethical issues. The study populations within these four cohorts are presented in Table 1. Briefly, TCGA largely had MIBC treated by cystectomy, GSE87304 had MIBC treated with neoadjuvant chemotherapy (NAC) prior to cystectomy, GSE48075 had a mix of NMIBC and MIBC treated with or without NAC and GSE32894 had transurethral resection of bladder tumor (TURBT).

### 2.2. Data Processing and Analysis

Bladder urothelial carcinoma Illumina Hi-Seq counts from TCGA were downloaded from the Genomic Data Commons (GDC) data portal, and corresponding clinical annotation including survival information was accessed via the TCGA Clinical Data Resource. Consensus MIBC classifications of TCGA cases were obtained from the consensus MIBC study. A comprehensive analysis using the edgeR package was performed to obtain the gene expression values [24].

### 2.3. Survival Analysis

Kaplan–Meier curves were used to determine the association between individual biomarkers (low vs. high expression) and prognosis. High expression was defined as >median, and low expression was defined as <median.

### 2.4. Univariate and Multivariate Analysis

The biomarkers associated with each multiplex test were evaluated by univariate Cox regression, and the relevant biomarkers were then evaluated using a multivariate Cox regression model to select the biomarkers that were most strongly associated with survival. All statistical analyses were performed using SPSS19.0.

### 2.5. KEGG Pathway Analysis

The Database for Annotation, Visualization and Integrated Discovery (DAVID) [25,26] was used to perform Gene Ontology (GO) functional analyses [27], reporting the top biological processes and cellular components.

## 3. Results

Using a series of gene expression datasets from TCGA and the GEO, we evaluated the association of the 10 biomarkers comprising the Oncuria™ diagnostic signature with prescribed bladder cancer molecular subtypes and with clinical outcomes. The TCGA cohort was parsed into luminal (*n* = 242, 59%) and basal (*n* = 166, 41%) subtypes based on gene expression profiles (Figure 1A) [19]. Analyses showed that 79% of samples in the luminal subtype showed high expression of VEGFA (*p* = 4.36 × 10^−8^ and SDC1 (1.62 × 10^−13^) (Figure 1B). Notably, tumors with a papillary morphology were significantly enriched in the luminal subtype (luminal 80% vs. basal 20%; *p* = 1.29 × 10^−4^). Conversely, 83% of samples in the basal subtypes had high expression of MMP9 (*p* = 1.49 × 10^−29^), MMP10 (*p* = 1.14 × 10^−2^), IL8 (*p* = 1.52 × 10^−8^), SERPINE1 (*p* = 2.7 × 10^−9^), APOE (*p* = 1.05 × 10^−10^) and SERPINA1 (*p* = 2.04 × 10^−23^) (Figure 1B). In addition, combined Oncuria™ signature is also significantly related to the luminal vs. basal subtype (*p* = 1.68 × 10^−^^4^) (Figure 1C). The basal subtype was enriched with tumors of a higher stage (T2-4, 93% vs. Ta and T1, 7.3%; *p* = 5.4 × 10^−34^).

We also tested whether the Oncuria™ biomarkers were differentially expressed with respect to a more contemporary consensus set [22] of six molecular classes of bladder cancer: luminal papillary, luminal non-specified, luminal unstable, stroma-rich, basal/squamous and neuroendocrine-like. Though there were limited subjects in some of the molecular classes (e.g., neuroendocrine-like and luminal non-specified), analyses showed that the Oncuria™ biomarkers could segregate samples into the six consensus subtypes (Appendix A). Together, these findings show that the expression patterns of Oncuria™ biomarkers are associated with reported molecular subtypes of bladder cancer.

Reported molecular subtypes have been reported to be associated with overall survival [17,18,19]. Here, the Kaplan–Meier analysis of the TCGA subjects indicated that high expression of the Oncuria™ signature was correlated with a significant reduction in overall survival (Figure 2; HR = 1.65; *p* = 0.000819). Analysis of each of the 10 individual Oncuria™ biomarkers revealed that high expression of MMP9 (HR = 1.36; *p* = 0.0395) and SERPINE1 (HR = 1.34; *p* = 0.0493) were associated with a significant reduction in overall survival, while high expression levels of VEGFA (HR = 0.66; *p* = 0.00574) were associated with a significant improvement in overall survival (Figure 3). Table 2 reports the Cox univariate and multivariate analysis, with VEGFA, MMP9, SERPINA1 and SERPINE1 being associated with survival probabilities.

Validation studies were performed using three independent, publicly available datasets (GSE87304, GSE48075 and GSE32894). In total, 9 of the 10 Oncuria™ biomarkers were present in each dataset. Notably, IL-8 was not present in GSE87304 and ANG was not present in GSE48075 and GSE32894. Similar to the analysis of TCGA data, we found that tumors with a relatively low expression of the combined Oncuria™ signature were associated with improved survival probabilities in all three datasets (GSE87304, recurrence-free survival probability, HR = 2.05 (1.19–3.52), *p* = 0.0118; GSE48075, overall-survival probability, HR = 1.98 (1.06–3.6), *p* = 0.0192; GSE32894, disease-specific survival probability, HR = 4.38 (2–9.6), *p* = 0.0012) (Figure 4A–C, respectively). In the GSE87304 cohort, only the low expression of MMP10 (HR = 0.607 (0.35–1.05), *p* = 0.0683) approached significance for improved recurrence-free survival (Appendix A). In the GSE48075 cohort, only the low expression of SERPINA1 (HR = 1.64 (0.912–2.95), *p* = 0.095) approached significance for improved overall survival (Appendix A). In the GSE32894 cohort, the low expression of APOE (HR = 2.67 (1.22–5.85), *p* = 0.0213), IL8 (HR = 4.21 (1.92–9.23), *p* = 0.00172), MMP9 (HR = 3.3 (1.51–7.23), *p* = 0.00669) and SERPINA1 (HR = 4.32 (1.97–9.47), *p* = 0.00134) and the high expression of SDC1 (HR = 0.158 (0.0719–0.347), *p* = 0.000099) and VEGFA (HR = 0.432 (0.197–0.946), *p* = 0.0432) were associated with improved disease-specific survival (Appendix A). Taken together, these findings validate the notion that multiplex signatures provide better prognostic models than individual biomarkers. Prospective validation in a larger cohort may lead to the derivation of a weighted algorithm that would maximize the utility of molecular signature(s) for subtyping and prognosis. Subsequently, GO enrichment analysis indicated that the biomarkers associated with the bladder cancer diagnostic signature were significantly enriched in the regulation of induction of positive chemotaxis and vascular permeability affecting the extracellular matrix, key processes in the growth of tumors (Appendix A).

## 4. Discussion

As well as our own work on a bladder cancer diagnostic test, several other groups have begun to identify panels of biomarkers for potential cancer detection. For example, through the analysis of nine gene promoters, Hoque et al. found that 69% of bladder cancer patients had methylation in at least one of four genes (CDKN2A, ARF, MGMT and GSTP1), whereas the control patients had no such methylation detectable [28]. By combining the data from all nine genes, a logistic prediction model was derived that achieved a sensitivity of 82% and specificity of 96%. Chung et al. selected 10 candidate hypermethylated genes from data collected from tumor tissue and monitored these in voided urine samples by quantitative, methylation-specific RT-PCR and identified a multigene predictive model comprised of five target genes (MYO3A, CA10, NKX6-2, DBC1 and SOX11) [29]. Sensitivity and specificity of this model were 85% and 95%, respectively. Further examples include RNA signatures proposed by Holyoake et al. [30], Hanke et al. [31] and Mengual et al. [32,33,34], with diagnostic sensitivities of 80–92% and specificities of 85–99%. Many of these studies have had a relatively small sample size, limited populations (i.e., few benign and confounding conditions included) and have not undergone extensive validation, but more comprehensive studies are ongoing.

Urothelial carcinoma is pathologically classified as non-muscle-invasive bladder cancer (NMIBC) or muscle-invasive bladder cancer (MIBC). The standard treatment for NMIBC is transurethral resection of bladder tumor (TURBT) for low-risk cases, or TURBT followed by intravesical therapy, such as BCG, for high-risk NMIBC, and the universal treatment for MIBC is radical cystectomy. A considerable number of NMIBC patients (50% to 80%) have tumor recurrence [35] and up to 45% progress to MIBC after 5 years, leading to poor survival rates associated with more advanced disease. Pathological staging is a key factor in current clinical decision making and prognosis of bladder cancer; nevertheless, the clinical outcomes of patients at the same stage often differ, indicating that the current staging system is not sufficient to reflect biological heterogeneity, and accurately determining the prognosis of patients is challenging. Prognostic evaluation models based on molecular signatures or subtypes may be able to better guide individualized treatment and improve outcome prediction.

Our analyses showed that the levels of biomarkers within the Oncuria™ diagnostic test could stratify patients into luminal (VEGFA (*p* = 4.36 × 10^−8^) and SDC1 (*p* = 1.62 × 10^−13^)) or basal (MMP9 (*p* = 1.49 × 10^−29^), MMP10 (*p* = 1.14 × 10^−2^), IL8 (*p* = 1.52 × 10^−2^), SERPINE1 (*p* = 2.7 × 10^−9^), APOE (*p* = 1.05 × 10^−10^) and SERPINA1 (*p* = 2.04 × 10^−23^)) subtypes (Figure 1). Furthermore, the survival curve analysis showed that multivariate models composed of specific Oncuria™ biomarkers (VEGFA, MMP9, SERPINA1 and SERPINE1) were associated with better outcomes (Table 2). In addition, discovery analysis using TCGA dataset found confirmed that multiplex signatures provided better prognostic models than individual biomarkers (Figure 2 and Figure 3). Validation analyses using three publicly available datasets (GSE87304, GSE48075, GSE32894) confirmed these findings (Figure 4, Appendix A). Furthermore, profiling mRNA levels of only 10 biomarkers of Oncuria™ predict overall, recurrence- and disease-free survivals, while comprehensive analyses will be required to classify other subtypes [18,19,22,23]. Lastly, the GO enrichment analysis indicates enriched biomarker activity within the extracellular space (Supplemental Appendix A). Our previous studies have demonstrated that the immunostaining patterns of the Oncuria™ biomarkers are enriched in human bladder stromal tissues in malignancy and associated with a reduction in overall survival [14]. As a pilot study, a number of limitations are evident. While the use of concise multiplex signatures for prognosis and subtyping is promising, as a retrospective study leveraging publicly available databases, interpretation requires caution. For example, the treatments that patients have received will influence outcomes and are highly heterogeneous and incomplete across cohorts; thus, we could not include this information in our analysis. Improving these aspects for future in-depth studies will enable comprehensive evaluation of the value of multiplex signatures in predicting clinical outcomes.

## 5. Conclusions

In summary, we have demonstrated that the biomarkers that comprise an established diagnostic signature have the potential to also have value for molecular subtyping and prediction of clinical outcomes for patients with bladder cancer. Specifically, patients with a high expression of the Oncuria™ signature were associated with a significant reduction in overall survival. Future work may enable the construction of algorithms that can define and weigh specific biomarkers for the stratification of patients for clinical management.

## Figures and Tables

**Figure 1 diagnostics-12-01801-f001:**
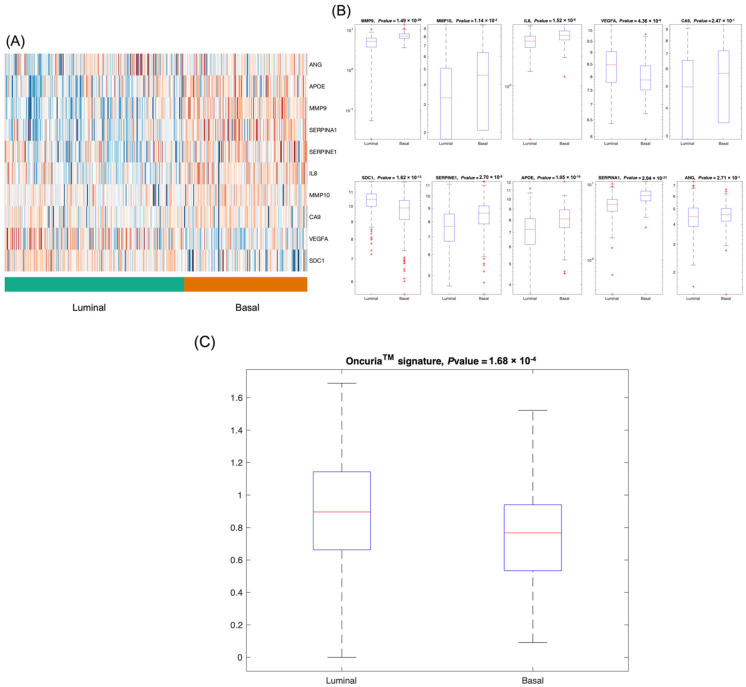
Correlation of the 10 biomarkers associated with Oncuria™ with luminal vs. basal tumors. (**A**) Heatmap illustrating application of each of the 10 biomarkers associated with Oncuria™ in stratifying luminal vs. basal tumors within the TCGA cohort. Blue to brown shows a trend from low to high gene expression. (**B**) Gene expression results of the individual biomarkers from the bladder cancer signature related to luminal vs. basal subtype. (**C**) Oncuria^TM^ combined signature is related to luminal vs. basal subtype.

**Figure 2 diagnostics-12-01801-f002:**
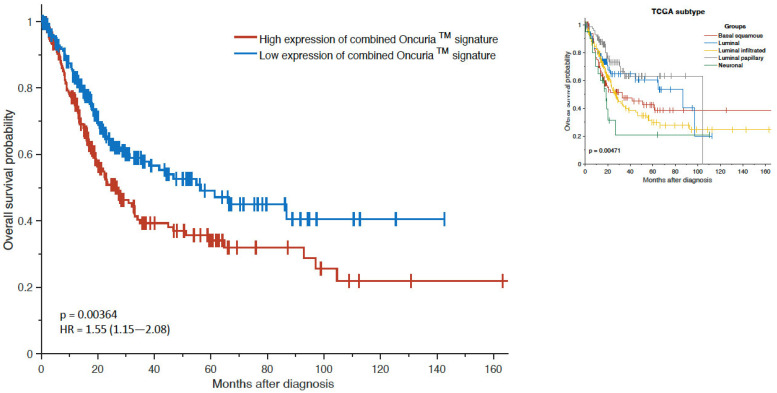
Kaplan–Meier survival curves for high vs. low expression of the combined Oncuria™ signature in TCGA cohort; insert depicts TCGA analyzed by the consensus model.

**Figure 3 diagnostics-12-01801-f003:**
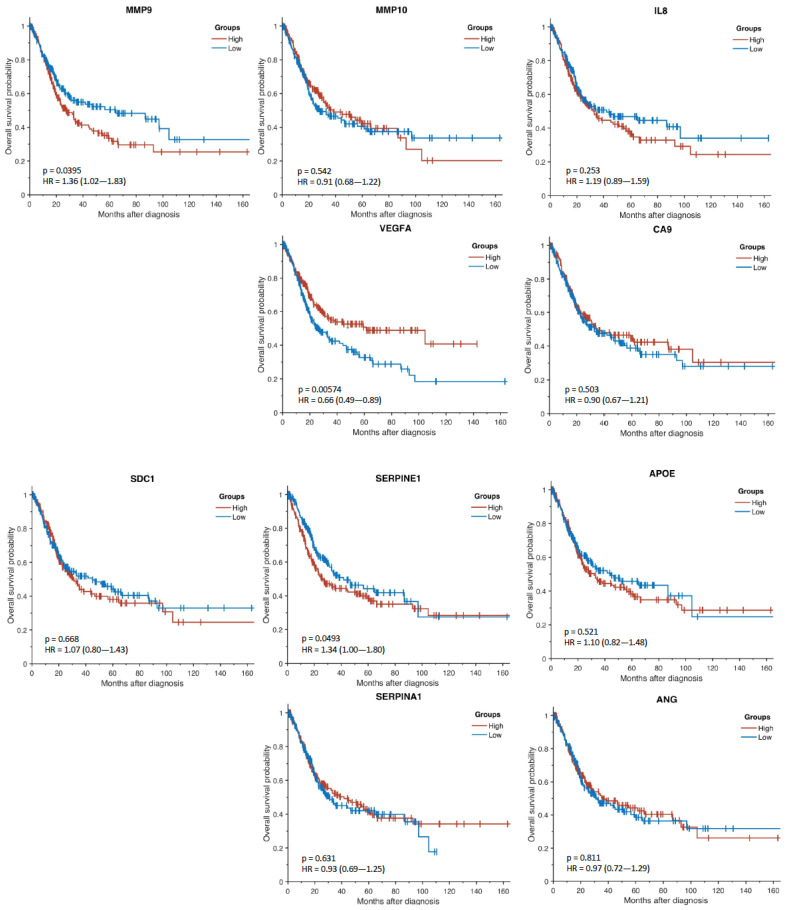
Kaplan–Meier survival curves for each of the 10 Oncuria™ biomarkers in the TCGA cohort.

**Figure 4 diagnostics-12-01801-f004:**
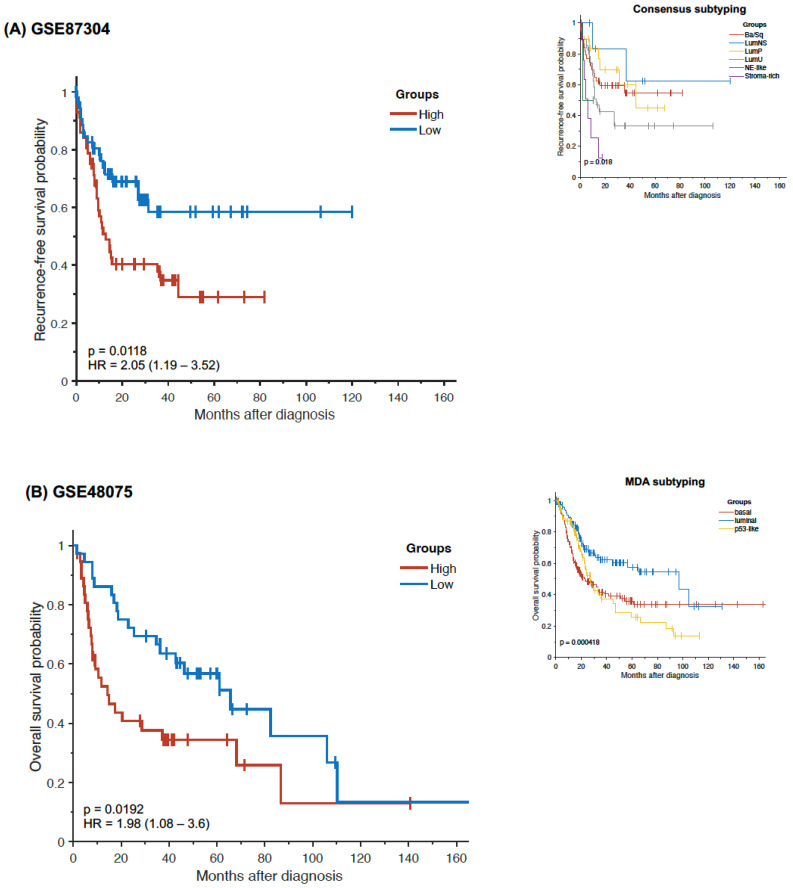
Kaplan–Meier survival curves for high vs. low expression of the combined Oncuria™ signature in (**A**) GSE87304 cohort; insert depicts GSE87304 analyzed by the consensus subtyping system, (**B**) GSE48075 cohort; insert depicts GSE48075 analyzed by the MDA subtyping system and (**C**) GSE32894 cohort; insert depicts GSE32894 analyzed by the model reported in the associated GSE32894 manuscript [18,19,23].

**Table 1 diagnostics-12-01801-t001:** Demographic and clinical-pathologic characteristics of study cohorts.

			TCGA		GSE87304		GSE48075		GSE32894
			(*N* = 412)		(*N* = 323)		(*N* = 73)		(*N* = 308)
Variable	Value	*n*	%	*n*	%	*n*	%	*n*	%
Age	<65	151	37.0	182.0	56.0	22.0	30.0	100	32.0
Age	≥65	261	63.0	136.0	42.0	51.0	70.0	208	68.0
Sex	Female	304	74.0	235.0	73.0	54.0	74.0	228	74.0
Sex	Male	108	26.0	88.0	27.0	19.0	26.0	80	26.0
Race	White	327	79.0	-	-	54.0	74.0	-	-
Race	Other	85	21.0	-	-	19.0	26.0	-	-
Stage	≤I	3	0.0	0.0	0.0	0.0	0.0	213	69.0
Stage	II	121	29.0	148.0	46.0	37.0	51.0	85	28.0
	III	196	48.0	123.0	38.0	16.0	22.0	7	2.0
Stage	IV	59	14.0	0.0	0.0	6.0	8.0	1	0.0
Grade	Low	21	94.0	-	-	-	-	153	50.0
	High	388	5.0	-	-	-	-	155	50.0

**Table 2 diagnostics-12-01801-t002:** Coefficients based on a Cox regression analysis of the 10 Oncuria™ biomarkers.

Variables	Univariate Analysis			Multivariate Analysis		
	HR	95% CI of HR	*p* Value	HR	95% CI of HR	*p* Value
SERPINA1	0.96	[0.82, 1.12]	0.57	0.76	[0.60, 0.95]	0.02
ANG	1.00	[0.87, 1.16]	0.97	1.01	[0.86, 1.17]	0.94
APOE	1.08	[0.93, 1.25]	0.34	0.94	[0.75, 1.17]	0.57
CA9	0.94	[0.82, 1.08]	0.39	1.03	[0.87, 1.21]	0.76
IL8	1.06	[0.91, 1.23]	0.46	1.06	[0.88, 1.27]	0.57
MMP9	1.21	[1.04, 1.41]	0.02	1.31	[1.04, 1.65]	0.02
MMP10	0.95	[0.82, 1.11]	0.54	0.92	[0.78, 1.08]	0.31
SERPINE1	1.15	[0.99, 1.33]	0.08	1.17	[0.98, 1.39]	0.08
SDC1	1.00	[0.86, 1.16]	0.95	1.01	[0.86, 1.19]	0.88
VEGFA	0.81	[0.69, 0.94]	0.01	0.79	[0.66, 0.94]	0.01

## Data Availability

TCGA-BLCA dataset is available at NCI Genomic Data Commons. GSE87304, GSE48075, and GSE32894 datasets are available at Gene Expression Omnibus (GEO) repository within the National Center for Biotechnology Information.

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
