# Peer review of "A Diagnostic Gene Expression Signature for Bladder Cancer Can Stratify Cases into Prescribed Molecular Subtypes and Predict Outcome"

_diagnostics, 2022, doi:10.3390/diagnostics12081801_

Round 1

Reviewer 1 Report

The current study analyzed a series of gene expression datasets from TCGA and the GEO to evaluate the potential utility of the biomarkers comprising the Oncuria™ test signature for the molecular subtyping of bladder cancer and the prediction of clinical outcome. The authors found that MMP9, MMP10, IL8, VEGFA, SDC1, SERPIN1, APOE, and SERPINA1 were significantly associated with either luminal or basal molecular subtypes and expression of combined Oncuria™ test signature could predicate the overall survial of bladder cancer patients. I have some comments for this study.

1. How did the authors difine the high or low expression of combined Oncuria™ test signature in the gene expression dataset from bladder patients. Although line 94-97 metioned high or low expression, it’s just for single gene.

2. How about using combined Oncuria™ test signature but not just single gene in distinguishing the luminal or basal molecular subtypes?

3. Expression of combined Oncuria™ test signature for predicating the overall survial of bladder cancer patients showed different outcomes in diffenent gene expression datasets. Please discuss and explain these inconsistent results in the discussion part.   

4. Patients from different gene expression platform showed great heterogeneity. Subgroup analysis of patients with homogeneity shoud be conducted.

Author Response

Reviewer 1

  1. How did the authors difine the high or low expression of combined Oncuria™ test signature in the gene expression dataset from bladder patients. Although line 94-97 metioned high or low expression, it’s just for single gene.

We first trained a cox regression model using the Oncuria™ expressions and patients’ survival time to decide the coefficients for each gene. Then, combined Oncuria™ signature is calculated by the sum of Oncuria™ expressions weighted by the coefficients. The high group is defined as combined Oncuria™ signature >= median value and the low group is combined Oncuria™ signature < median value.

  1. How about using combined Oncuria™ test signature but not just single gene in distinguishing the luminal or basal molecular subtypes?

Thank you for the suggestion, it makes sense. We generated the box-plot illustrating that Oncuria™ signature is related to luminal vs. basal subtype and added it in Fig 1 (C).

  1. Expression of combined Oncuria™ test signature for predicating the overall survial of bladder cancer patients showed different outcomes in diffenent gene expression datasets. Please discuss and explain these inconsistent results in the discussion part.

Because the cohort is different, the results could be different. However, please note that the trend was consistent in all cohorts, specifically the patients with high expression of 10 biomarkers showed worse survival as compared to the patients with Low expression of 10 biomarkers.

For cohorts, TCGA largely had MIBC treated by cystectomy, GSE87304 had muscle invasive bladder cancer (MIBC) treated with neoadjuvant chemotherapy (NAC) prior to cystectomy, GSE48075 had a mix of NMIBC and MIBC treated with or without NAC and GSE32894 had transurethral resection of bladder tumor (TURBT). The descriptions of cohort has been yellow highlighted (Line 79-82).

  1. Patients from different gene expression platform showed great heterogeneity. Subgroup analysis of patients with homogeneity shoud be conducted.

First, we don’t think the different gene expression platforms effect the analysis. We have analyzed each dataset (cohort) individually and so we have not merged the datasets. In addition, we didn’t compare the results between datasets. Second, each cohort data was shown with each inert showing different subtypes. The each subtype was used in the original paper that introduce the dataset (i.e. TCGA, GSE87304, GSE48075, and GSE32894 papers used TCGA subtypes, 6 consensus subtypes, MDA subtypes, and Sweden subtypes, respectively). Thus, we individually compared the performance of our 10 biomarkers with their subtypes.

Reviewer 2 Report

The manuscript is well conceived. I recommend it for publication.

Author Response

Reviewer 2

The manuscript is well conceived. I recommend it for publication.

Thank you very much.

Reviewer 3 Report

Dear researchers,

I would like to congratulate you on highly interesting work. Nevertheless, in its current form it cannot be published. ‘Discussion’ part is written in a poor way. I recommend to confront your own results with other researchers’ data. Lines 226-238 are not a discussion but they only present results.

Good luck !

Author Response

Reviewer 3

I would like to congratulate you on highly interesting work. Nevertheless, in its current form it cannot be published. ‘Discussion’ part is written in a poor way. I recommend to confront your own results with other researchers’ data. Lines 226-238 are not a discussion but they only present results.

Thank you very much for the suggestion. We have revised discussion section based on your suggestions.